# IMAGE SCORE: HOW TO SELECT USEFUL SAMPLES

## ABSTRACT

There has long been debates on how we could interpret neural networks and understand the decisions our models make. Specifically, why deep neural networks tend to be error-prone when dealing with samples that output low softmax scores. We present an efficient approach to measure the confidence of decision-making steps by statistically investigating each unit's contribution to that decision. Instead of focusing on how the models react on datasets, we study the datasets themselves given a pre-trained model. Our approach is capable of assigning a score to each sample within a dataset that measures the frequency of occurrence of that sample's chain of activation. We demonstrate with experiments that our method could select useful samples to improve deep neural networks in a semi-supervised leaning setting.

## 1 INTRODUCTION

Since its development two decades ago, Convolutional Neural Networks (LeCun et al., 1998) gradually dominate in tasks such as image classification and object detection. Various network architectures (Krizhevsky et al., 2012; Simonyan & Zisserman, 2014; He et al., 2016) all prove the power of CNNs. On the other hand, many people believe that the end-to-end training approach makes the network a black box. However, recent efforts in visualizing (Zeiler & Fergus, 2014; Mahendran & Vedaldi, 2015) and understanding (Bau et al., 2017; Koh & Liang, 2017) CNNs reveal that despite the large number of hidden parameters, convolutional networks show intrinsic patterns during training.

The idea of explainable artificial intelligence suggests that in order to further improve deep models, we should first be able to understand the intrinsic structures that the models learned from training, i.e. why the model classify a certain image to a specific class, and how did the model make that decision. Recently more researchers focus on the aforementioned problem and try to interpret CNNs. The network dissection method, proposed by Bau et al., is an efficient way to score unit interpretability in terms of trained CNNs operating on Broden dataset. The method considers each image's activation map and defines an intersection over union score to quantify each concept's interpretability. In another paper (Zhou et al., 2014), it is suggested that each individual unit behaves as detectors in a network trained to classify scenes. There are also papers that employ inverse method to study networks (Nguyen et al., 2016), where the authors suggest that instead of trying to interpret CNNs, we should synthesize images base on the models, and interpretability of networks could be judged by the quality of synthesized images. Beyond simply analyzing convolutional neural networks, there are also models designed to learn a better representation of semantic meanings, which are encoded by specificity and generality of neurons in each layer (Yosinski et al., 2014). Dynamic routing between capsules (Sabour et al., 2017) could parse images into capsules where each capsule encodes a specific semantic meaning. Zhang et al. proposed an interpretable convolutional neural network where high convolutional layers represent certain object parts (Zhang et al., 2018).

However, almost all of the existed works investigate only the models and ignore the relationship between models and samples. The network dissection method (Bau et al., 2017) focuses on the amount of units that could be interpreted, while the neural activation constellation method (Simon & Rodner, 2015) could learn part models in an unsupervised manner. Different than the above-mentioned approached, we believe that sampling distributions, i.e. datasets, encode their own features; and that no network would work on all distributions. As a consequence, relationships between models and samples should examined more carefully.

Unlike traditional supervised learning that requires fully labeled data to train a classifier, semi-supervised learning requires only a portion of data to be labeled, which demands less effort in data preparation and is more practical in real life. Consider a semi-supervised leaning task where both the labeled and the unlabeled data come from the same sampling distribution. A straightforward technique is that we first train a model on the labeled data, and then use this model to classify the unlabeled parts of the dataset to enlarge the training set. After this, all the data are labeled regardless of correctness. The model could then be re-trained by iteratively performing the above steps. However, experiments have shown that this approach does not work well on CNNs (see section 3 for details). The reason is two-fold:

- Parts of the data are labeled by our model correctly with extreme confidence, i.e. probability generated by softmax layer is close to 1. This could happen only if our model already learned the features during training, therefore adding these data into the training set will not provide any improvement.

- The model does not pick up some of the features that are contained in the unlabeled data during training. Therefore, those data are doomed to be labeled incorrectly and could potentially weaken our model upon added into the training set.

To summarize these two problems: could we use a particular sample to improve our model; and could we trust our model to label a particular sample correctly. We propose a method to automatically assign scores to data that represents both confidence (could we trust it) and interpretability (could we use it) based on a pre-trained network. Our approach mainly focuses on activations of individual neurons within a network. For a pre-trained CNN and some images within one category, we focus on images' activation maps generated by all of the convolutional and linear layers. The activation maps contain information about each image's relationship with the activated neurons. Then our method could identify neurons that are commonly activated and thereby assigning scores to each individual image based on it's chain of activation. Samples with high scores are assumed to be interpretable and experiments have shown that they are also trustworthy. This entire procedure is unsupervised, i.e. our approach does not need label information to assign image scores. Therefore, we could apply our method to semi-supervised learning tasks including image classification and object detection.

Structure of this paper is the following: in section 2 we formally define image scores and then examine some key properties of our approach; in section 3 we conduct experiments in a semi-supervised learning setting to demonstrate how scoring the images could help with improving a pre-trained model, experiments are done in image classification and object detection tasks; section 4 concludes the entire work.

## 2 APPROACH

### 2.1 INTUITION

Given a pre-trained model and a dataset, we could say that the dataset consists of "good" and "bad" images in the sense of interpretability (Figure 1). While filters in good images could learn object themselves, filters in bad images would easily be confused with image background information, as shown in Figure 1 (second row; first, third, fourth, sixth, and seventh from the left). It is also possible that filters in bad images could not pick up feasible features at all, like these in Figure 1 (second row; second, and fifth from the left). In either of these two cases, we say the image could not be interpreted by the current model, and thus we would assign a relatively low score. By this definition of interpretability, if an image is assigned a high score, it's chain of activation should include little to no background information (see Figure 1; fourth row); this makes the connection to confidence in that these images should be classified correctly.

Based on the principles above, we propose the image score method to measure interpretability of individual images give a pre-trained model. The intuition is simple: all correctly classified images should have similar chain of activation, while incorrectly classified images should have very different activations both within themselves and with correctly classified images.

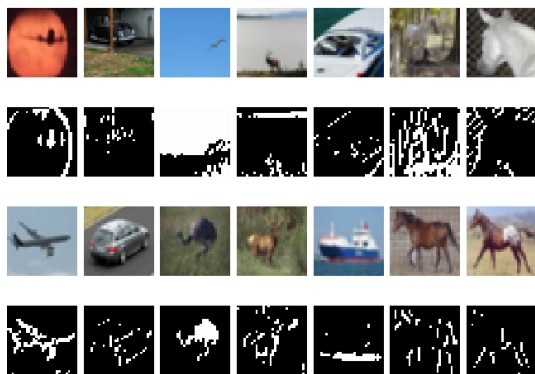

Figure 1: From top to bottom: bad images, bad features, good images, good features. Features are feature maps generated by one filter in the first convolutional layer of a pretrained CNN. Our method assigns scores to automatically differentiate good and bad images.

## 2.2 IMAGE SCORE

Consider image set $\mathcal{I}$ that contains all the images in one specific class. Denote $I \in \mathcal{I}$ an image.

For each image $I$ and layer $l$ in a pre-trained CNN, we have a corresponding activation map $X^{I,l}$. By saying layer, we mean a collection of functions that consist of a convolution function for convolutional layers or a linear function for classification layers, (potentially) a normalization function, an activation function, and (potentially) a pooling function. By saying activation map, we mean the tensor generated by a specific layer.

Let $X^l = \sum_{I \in \mathcal{I}} X^{I,l}$ and threshold $t^l = \max(X^l)/2$. The idea here is that if a neuron is commonly activated when processing a class of images $\mathcal{I}$, sum of activation values across $\mathcal{I}$ should be large in comparison with neurons that are not commonly activated. We choose the threshold to be related to $L_\infty$ norm, but this could also be done with other norms such as $L_1$ and $L_2$.

Rather than the activation values, we care more about whether a specific neuron is activated by some images at all. This is because after the softmax layer, we pick the index of the maximum element to be our prediction, and it doesn't matter whether this maximum value is $1$ or $0.101$ as long as it is the largest. We create a binary mask of activation to reflect this. Denote $M^l$ the "correct" activation mask of layer $l$, and each element $M_i^l$ of $M^l$ satisfies $M_i^l = \mathbf{1}\{X_i^l \geq t^l\}$, i.e. elements in binary mask $M^l$ is $1$ if and only if the corresponding activation value in $X^l$ is greater than the threshold. Note that $M^l$ is a binary mask for all the images $I \in \mathcal{I}$ and it contains the information of "important" activations. Similarly, denote $M^{I,l}$ the activation mask for image $I$ in layer $l$ and $M_i^{I,l} = \mathbf{1}\{X_i^{I,l} \geq t^l\}$. As mentioned earlier, the intuition is that if some images are labeled correctly, they are classified accurately in a similar fashion. The rest of the job is simply to compare each image's activation mask $M^{I,l}$ and the "correct" activation mask $M^l$ we constructed.

For each image $I$, define score $s^I$ as

$$s^I = \sum_l \log \left( \frac{\sum_i M_i^{I,l}}{\sum_i M_i^l} \right)$$

The image score $s^I$ we defined represents similarity between an image's chain of activation and the "correct" chain of activation. The "ground truth" that we established in turn reflects common activations among all images.

## 2.3 PROPERTIES OF IMAGE SCORES

Convolutional neural networks are susceptible to bias such that even if the model achieve good classification performance, it still encodes biased information. More often than not, a CNN might extract image background information to perfect itself, i.e. it may identify cloud as a feature when

classifying airplanes. The encoded biased information is beneficial during training but problematic during testing since the biased information in training set and testing set might not be the same. As a matter of fact, this is why if we test on the training set, performance will supersede actual accuracy significantly, as shown in Figure 2 (left). Yet regardless of dataset bias, both training accuracy and testing accuracy increase as image score increases. On the other hand, observe that the number of training and testing images fall into any interval of image scores do not show major difference.

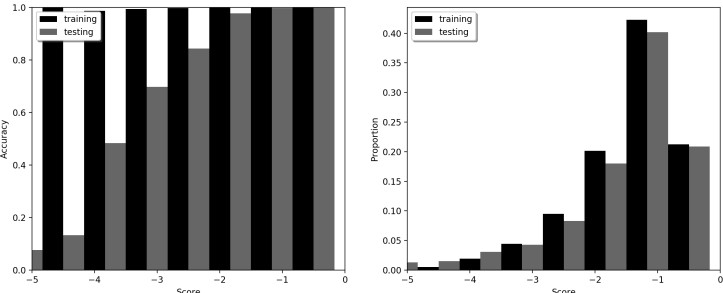

Figure 2: Scores of images in CIFAR-10 (class 0) dataset, pre-trained 50 epochs on VGG16. Class testing accuracy is $90.4\%$. Vertical axis in the left image represents labeling accuracy; vertical axis in the right image represents proportion of images falls into a certain interval of image score. Horizontal axis represents intervals of image scores. Notice that bars are grouped together, i.e. the rightmost two bars represent testing images falling into score interval $(-0.7, 0.0]$ and training images falling into the same interval, respectively.

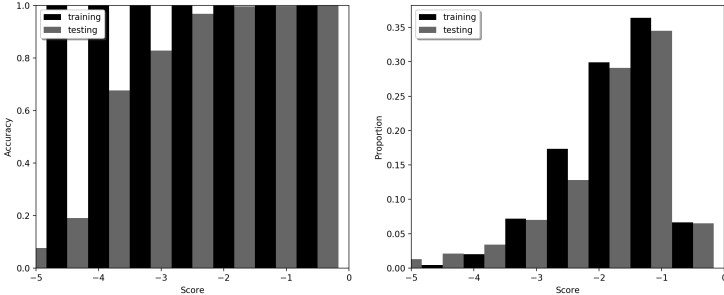

Figure 3: Scores of images in CIFAR-10 (class 0) dataset, pre-trained 500 epochs on VGG16. Class testing accuracy is $90.9\%$.

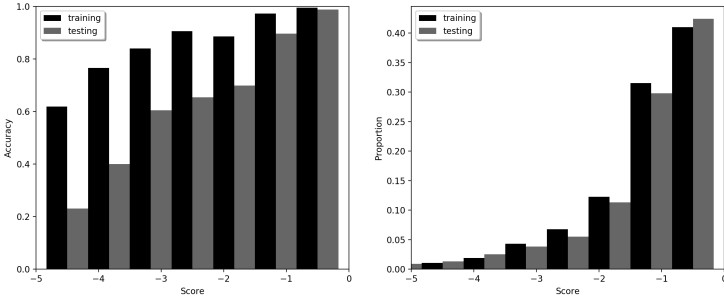

Figure 4: Scores of images in CIFAR-10 (class 0) dataset, pre-trained 10 epochs on VGG16. Class testing accuracy is $83.7\%$.

If we consider our model to be a classifier, as training proceeds more and more bias information will be introduced, making classification more fine-grained. As a result, most of the images will have minor differences in activation maps since we expect bias of each image to be different. Therefore,

we expect to see less images to have very high scores, as illustrated in Figure 2-4. In the meantime we would expect images in all intervals to gain accuracy until the loss function convergences, which is also revealed in the above figures.

# 3 EXPERIMENTS

## 3.1 IMAGE CLASSIFICATION

In the last section we conclude that throughout training phase, labeling accuracy of images with high scores increases to nearly $100\%$. This provides us theoretical foundations for semi-supervised deep learning tasks.

The intuition is that since we could achieve nearly $100\%$ labeling accuracy for images with high scores, we could simply treat the labels generated by our model as ground truth and add the labeled images back into the training set for re-training. In section 1 we discussed that the output of softmax layer is a probability distribution that represents likelihood of a certain image belongs to a certain class. In this section we demonstrate that if we choose samples base only on that distribution, there will not be any improvement to the model. However, if we choose samples base on image score, labeling accuracy will show statistically significant improvement.

**Dataset**. There are two experiments done on CIFAR-10 dataset (Krizhevsky & Hinton, 2009). The dataset contains 50,000 training images and 10,000 testing images; all of the data are labeled. There are a total of 10 classes, where each class has exactly 5,000 training images and 1,000 testing images. To apply semi-supervised learning, we manually separate the training dataset into two parts: one part contains 30,000 images that we consider as labeled, and the other part contains 20,000 images that we consider as unlabeled. We call the first part training set and the second part additional dataset.

**Two baselines**. We establish the first baseline by training on the training set and hope that we could outperform this baseline by investigating the additional dataset. As discussed, utilizing output from softmax layer is another approach to semi-supervised learning. We construct a second baseline, which we call the softmax baseline. Our model is first trained on the training set and then use the trained model to label the additional dataset. We then select all the images in the additional dataset that we categorized as class $c$ and observe the likelihood that each image actually belongs to class $c$ base on the probability distribution generated by the softmax layer. Then we choose some images with high softmax score and put those images, together with the generated labels, into the training set for re-training. The number of images we choose is $\alpha_c \times l_a$, where $l_a = 20,000$ is the size of additional dataset and $\alpha_c$ is a preselected constant, potentially different for each class $c$.

**Training**. We perform semi-supervised learning in a similar manner as we establish the softmax baseline. We first train a model on the training set and use this trained model to label the additional dataset. Then we choose all the images in the additional dataset that we labeled as class $c$, and denote this image set $\mathcal{I}^{a,c}$. Denote $\mathcal{I}^{t,c}$ all the images in training set that actually belongs to class $c$. Note that $\mathcal{I}^{a,c}$ is error-prone while $\mathcal{I}^{t,c}$ is not because unlike the training set, we don't have label information about the additional dataset. We could then use $\mathcal{I}^{t,c}$ to build the "correct" activations for each layer as described earlier and use the activation maps to score images in $\mathcal{I}^{a,c}$. After this we choose some images with high scores and put them into the training set for re-training, just like when we build the softmax baseline. Here the number of images chosen is $\alpha_c \times l_t$ where $l_a = 20,000$ is the size of additional dataset and $\alpha_c$ is a preselected constant.

**Implementation details**. We use two different models and techniques to illustrate that our approach is universal. The first experiment is done on VGG16 (Simonyan & Zisserman, 2014). In this experiment we choose $\alpha_1 = 0.5$ and $\alpha_c = 0.3$ for every $c \neq 1$. The second experiment is done on ResNet18 (He et al., 2016). During training we apply batch normalization (Ioffe & Szegedy, 2015) and dropout (Srivastava et al., 2014) techniques. In this experiment we choose $\alpha_c$=[0.2, 0.3, 0.2, 0.1, 0.2, 0.2, 0.3, 0.2, 0.4, 0.2] where $c = 0...9$, respectively. In all of the experiments, we use cross entropy loss and Adam (Kingma & Ba, 2014) as optimizer. The models are first trained on the training set for 100 epochs, and then perform either baseline training, softmax baseline training, or semi-supervised training for another 100 epochs. Note that we could control the time gap between enlarging the training set, in this experiment the training set is updated once every epoch.

**Data processing**. Since baseline classification accuracy is already high, we apply statistical significance tests to determine if there's any actual improvement to the network. In Table 1-2, wherever it says "difference", we actually suggest a difference in mean bootstrapping test. The following column of p-values are generated by one-sided tests. We choose significance level to be $\alpha = 0.05$, i.e. one-sided $95\%$ confidence interval could be constructed by the bootstrapping test.

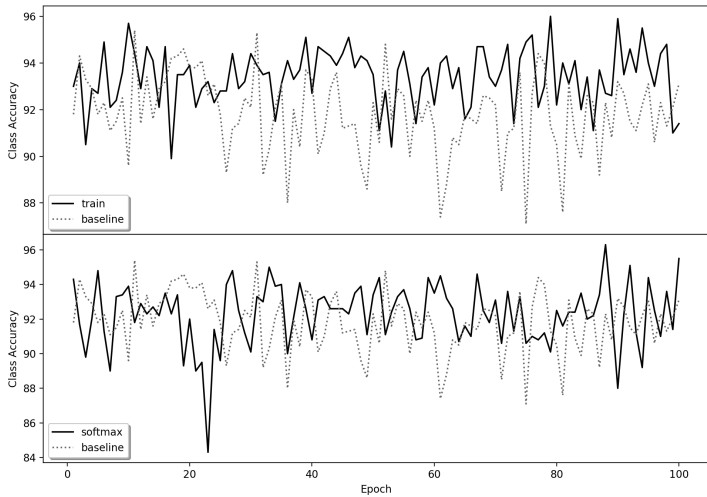

Figure 5: Labeling accuracy of CIFAR-10 (class 0) dataset. The model is pre-trained on ResNet18 for 100 epochs and then re-trained for another 100 epochs via semi-supervised learning (top) and softmax baseline training.

Table 1: Statistics of all ten classes of images in CIFAR-10 dataset, trained on VGG16. Column1 is class number; column2 is baseline accuracy; column3 is semi-supervised training accuracy; column4 is the difference between column3 and column2; column5 is significance test of column4; column6 is softmax baseline accuracy; column7 is the difference between column6 and column2; column8 is the significance test of column7; column9 is the difference between column3 and column6.

|         | baseline | train  | diff_tb | p-value_tb | softmax | diff_sb | p-value_sb | diff_ts |
|---------|----------|--------|---------|------------|---------|---------|------------|---------|
| Class 0 | 85.800   | 87.200 | 1.399   | 0.0000     | 86.055  | 0.253   | 0.2380     | 1.142   |
| Class 1 | 91.542   | 92.475 | 0.935   | 0.0037     | 92.039  | 0.499   | 0.0690     | 0.436   |
| Class 2 | 74.422   | 78.288 | 3.873   | 0.0000     | 75.205  | 0.785   | 0.0650     | 3.083   |
| Class 3 | 68.941   | 72.450 | 3.614   | 0.0000     | 69.669  | 0.725   | 0.1160     | 2.788   |
| Class 4 | 82.605   | 84.458 | 1.850   | 0.0002     | 82.513  | -0.094  | 0.5720     | 1.944   |
| Class 5 | 75.318   | 77.342 | 2.032   | 0.0004     | 76.744  | 1.429   | 0.0058     | 0.609   |
| Class 6 | 86.987   | 87.944 | 0.955   | 0.0008     | 87.453  | 0.464   | 0.1010     | 0.490   |
| Class 7 | 86.690   | 87.439 | 0.746   | 0.0270     | 86.699  | 0.009   | 0.4880     | 0.742   |
| Class 8 | 89.235   | 90.194 | 0.963   | 0.0002     | 89.970  | 0.729   | 0.0098     | 0.221   |
| Class 9 | 89.369   | 90.396 | 1.023   | 0.0022     | 89.678  | 0.304   | 0.2060     | 0.721   |

**Data analyze**. Even without statistical significance tests, from Figure 5 we could observe that the softmax baseline training doesn't perfect our model while the semi-supervised training improves the network. For VGG16, p-values reveal that when applying image scores, all the classes show significant improvement than the baseline on a significance level of $\alpha = 0.05$, while only class 5 and class 8 suggest major perfection of the network when using softmax scores to select samples. Similar circumstances occur when re-training a ResNet18 model: semi-supervised re-training resulted in 9 out of 10 classes improved; while the softmax baseline training method only improved 3 of the total 10 classes. From Table 3-4 we could also see that p-value is smaller for ResNet18 network, this is natural because the additional two convolutional layers provided by ResNet18 make the model more

Table 2: Statistics of all ten classes of images in CIFAR-10 dataset, trained on ResNet18.

|         | baseline | train  | diff_tb | p-value_tb | softmax | diff_sb | p-value_sb | diff_ts |
|---------|----------|--------|---------|------------|---------|---------|------------|---------|
| Class 0 | 91.830   | 93.404 | 1.573   | 0.0000     | 92.277  | 0.455   | 0.0330     | 1.126   |
| Class 1 | 95.638   | 96.515 | 0.878   | 0.0000     | 95.631  | -0.005  | 0.5150     | 0.884   |
| Class 2 | 86.274   | 88.000 | 1.731   | 0.0000     | 85.516  | -0.756  | 0.9950     | 2.484   |
| Class 3 | 80.371   | 81.831 | 1.457   | 0.0000     | 80.696  | 0.325   | 0.2210     | 1.131   |
| Class 4 | 91.819   | 93.279 | 1.460   | 0.0000     | 91.433  | -0.383  | 0.9660     | 1.844   |
| Class 5 | 85.831   | 88.438 | 2.609   | 0.0000     | 86.647  | 0.817   | 0.0130     | 1.793   |
| Class 6 | 93.435   | 94.571 | 1.137   | 0.0000     | 93.826  | 0.392   | 0.0300     | 0.741   |
| Class 7 | 92.487   | 93.313 | 0.824   | 0.0000     | 92.617  | 0.133   | 0.2460     | 0.695   |
| Class 8 | 94.377   | 95.827 | 1.450   | 0.0000     | 94.188  | -0.186  | 0.8750     | 1.636   |
| Class 9 | 93.958   | 94.260 | 0.306   | 0.0820     | 93.973  | 0.014   | 0.4810     | 0.286   |

error-endurance. This means that small error or bias will not affect our estimation to the "correct" activation significantly.

**Model Assembly**. Our previous discussions involve training class-by-class. It is also possible to assemble those trained networks with boosted labeling accuracy on different classes together in order to enhance performance on the entire dataset. Suppose we have 10 models with boosted performance on 10 different classes. Suppose model $M_k$ focus on class $k$, i.e. $M_k$ is trained to categorize class $k$ images with better accuracy. Then we use $M_k$ to label all the test images and keep track of the images that are categorized as class $k$; we also record the image score assigned to an image if it is labeled as class $k$. We do this for all the models. After this step, some of the images will be labeled multiple times. We label those images based on which model assigns the highest score. From Table 3 we could see that performance of our assembled model supersedes the baseline accuracy by about $1\%$.

Table 3: Labeling accuracy of the assembled model. Semi-supervised training technique is applied on VGG16 network.

| epoch    | 10    | 20    | 30    | 40    | 50    |
|----------|-------|-------|-------|-------|-------|
| baseline | 82.78 | 82.57 | 82.79 | 82.85 | 82.9  |
| train    | 83.65 | 84.37 | 84.04 | 84.34 | 83.69 |

### 3.2 OBJECT DETECTION

Besides image categorization, object detection is another well-studied topic in computer vision. Since the development of region-bases CNN (Girshick et al., 2014), we now have better and faster techniques such as (Girshick, 2015; Ren et al., 2015). In the previous sections we discussed how to interpret convolutional neural networks and how to use interpretability to automatically select samples in order to apply semi-supervised learning techniques. Since the foundation of faster R-CNN (Ren et al., 2015) is still VGG and ResNet, we transfer our attention to object detection task and investigate an end-to-end semi-supervised training technique without having to do per-class training first.

**Dataset**. In this section, we apply faster R-CNN on Pascal VOC datasets. Both the VOC2007 and VOC2012 datasets have similar forms: each image in the dataset has ground truth bounding boxes that denot objects, and each object is associated with a label that belongs to one of the twenty categories. The only difference is that VOC2007 has a labeled testing set, while testing set of VOC2012 is unlabeled. Therefore, it is natural to treat VOC2007 as training set and VOC2012 as additional dataset, together we could apply semi-supervised learning techniques.

**Baselines**. In most of the papers examining pascal datasets there are two baselines. The first baseline is that models are trained solely on VOC2007 training set and tested on VOC2007 testing set. The second baseline is that models are trained on VOC2007+VOC2012 training sets and tested on

VOC2007 testing set. Since we are only utilizing parts of information from VOC2012 to do semi-supervised learning, the assumption is that our final accuracy lays somewhere in between these baselines.

**Implementation**. Implementation in this experiment is essentially the same as the one used in image categorization. Our approach only needs a feed forward step to calculate "correct" activation, image score, etc. Therefore, no matter how difficult a task is, as long as the basic architecture of the convolutional neural network remains the same, modifications to adapt the task is minimal. The only difference in this experiment is that instead of building activation maps for each class, we build a general "correct" activation for the entire dataset. Consequently, instead of selecting images base on which class it is assigned to, we simply select images with the highest scores. In this experiment we use an ImageNet pre-trained VGG16 network, and train the model on VOC2007 for 60,000 iterations, where each iteration only process one image. Then we use this trained network and the training set of VOC2007 to build "correct" activation. After this we use the constructed activation mask to score all the images in VOC2012 training set, and select images with top $30\%$ score. After all ground truth labels of selected images are replaced by predicted labels, we put those images into our training set for re-training. The re-training process lasts for another 10,000 iterations. Note that we only update the training set once during the entire training process.

Table 4: Mean AP of faster R-CNN model. The model is a VGG16 that is already trained for 60,000 iterations. Here "train" stands for semi-supervised training that utilizes part of VOC2012, and "baseline" stands for model only uses VOC2007 training set.

| iter | 61000 | 62000 | 63000 | 64000 | 65000 |
|---|---|---|---|---|---|
| train | 0.713 | 0.715 | 0.711 | 0.718 | 0.719 |
| baseline | 0.706 | 0.706 | 0.705 | 0.710 | 0.707 |
| iter | 66000 | 67000 | 68000 | 69000 | 70000 |
| train | 0.717 | 0.716 | 0.715 | 0.716 | 0.719 |
| baseline | 0.709 | 0.710 | 0.711 | 0.709 | 0.707 |

From Table 4 we could observe that our semi-supervised training method increases mAP about $1\%$. On the other hand, a model trained on both VOC2007 and VOC2012 training sets yields mAP 0.742, which is well above both our approach and the baseline. This is exactly what we expected. Our approach is above the baseline, and since we only use $30\%$ of (potentially incorrectly labeled) data from VOC2012 training set, it is plausible that the image score method is not comparable to the ideal model.

## 4 CONCLUSION

With growing interests in interpreting convolutional neural networks, we propose an alternative measure. Instead of focusing on the models, we focus on the interactions between models and data. There are two problems mentioned in section 1 when applying semi-supervised leaning techniques: some data are useless since their features were already learned; and some data are labeled incorrectly which could weaken the model. Our image score approach could efficiently select samples that are proven to be useful in a forward pass. The two mentioned problems could be resolved simultaneously by our approach; and the image score method has shown statistically significant improvements in comparison with the softmax baseline.

Semi-supervised leaning has proven to be efficient in multiple scenarios. However, when applying to deep learning, the traditional and naive way does not work well. Unlike softmax score which only encodes confidence, image score represents both interpretability and confidence. We could therefore select explainable and trustworthy samples from additional datasets to apply semi-supervised learning methods.

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
