# OpenReview forum: "Image Score: how to select useful samples"
_ICLR.cc/2019/Conference_

### Official Review · AnonReviewer2 · 2018-11-02
**A semi-supervised learning paper, lacking thorough experimentation**

**Rating:** 3
**Confidence:** 3

**Review:**

This paper proposes a new metric called the image score that compares the similarity of activation between a given image with a pool of groundtruth images. The paper finds it useful for semi-supervised learning with self-teaching, where the network picks the most confident sample and use the network prediction as the label. It finds that the proposed method is better than 1) not using the unlabeled data and 2) using softmax as an indicator for model prediction certainty.

Motivation: The introduction begins by motivating the interpretability story of deep learning, but I don’t see gaining any more interpretability by reading the rest of the paper. The paper proposes to improve interpretability by assigning a score to each individual example, but then the obtained scores are not properly analyzed in the paper, and only final classification accuracy is evaluated. What are the training samples that makes the model make certain decision at test time? How to measure the correlation between the usefulness of training samples and the proposed image score? These questions left unanswered in the paper. Figure 1 helps a little bit, but then the top row is not necessarily the bad images, but maybe hard examples that needs extra attention to learn. Therefore, I think the end results presented in the experiments do not align with the motivation. Rather than shooting for interpretability, this is just another semi-supervised learning paper.

Models: The major issue of this paper is the model formulation that is not well motivated. The intuition of how the authors come up with the equation for computing the image score is not well explained. Hence the formulation seems very ad-hoc, and it is unclear why this is the selected method.

Experiments: As a semi-supervised learning paper, a common setting for CIFAR-10 is to use 4k labeled images. Here, the method uses 30k, which is 7.5x the size of the usual setting. It also does not compare to prior semi-supervised learning work (e.g. one of the recent one is: https://arxiv.org/abs/1711.00258). The only two baselines discussed here are weak. Also the improvement from the baselines by using the proposed method is not very significant.

Comparison: Figure 2-4 shows some positive correlation between the accuracy and score, which is fair, but it doesn’t compare to any baselines--the only one we have is softmax baseline and it is not shown in the figure.

In conclusion, I couldn’t see how the paper improves interpretability as claimed in the introduction. The proposed method seems ad-hoc, without any justification. Being considered as a semi-supervised learning paper, it lack significant amount of comparison to prior work and adopting a common semi-supervised benchmark. Due to the above reasons, I recommend reject.

---
Minor points:
“...almost all of the existed works investigate only the models and ignore the relationship between models and samples”. This is over-exaggerated. I believe most of the visualization techniques are dependent on the actual input samples. It is true to say about “training samples” not “samples” in general.

“all correctly classified images should have similar chain of activation, while incorrectly classified images should have very different activations both within themselves and with correctly classified images”. This claim seems not backed up. How do you know it is the case for “all” correctly classified images? What defines similar/different?

---

### Official Review · AnonReviewer1 · 2018-11-04
**Not likely to be significant enough**

**Rating:** 4
**Confidence:** 4

**Review:**

This paper proposes image score, to use the amount of strong activations in each single image compared with the average activation on the entire dataset as a metric of how well the image is interpreted by a particular deep model. I am not sure whether that intuition makes sense, and there seem to be a lot of ways the simplistic equation can be broken. Section 2.3 shows some intuition of a higher score corresponding to higher testing accuracy, but somehow is done only on 1 class in CIFAR and I don't think that is generalizable to other classes and especially other classification problems.

The paper claims interpretability but I don't see any experiments verifying interpretability. The experiments are done on a semi-supervised learning task, where the "image score" is used to select some unlabeled examples as labeled by trusting a partially trained classifier. Hence we would have to evaluate it as a semi-supervised deep learning paper. Note that the amount of labeled examples in this paper is significantly higher than most semi-supervised approaches, which leaves the question that if extremely few supervised examples have been used, whether this approach will already fail. Although the model showed some improvements over the baseline, there has been no comparison at all with any existing semi-supervised deep learning approaches. Any semi-supervised learning approach usually outperforms the supervised baseline (used in this paper) by a bit, so I don't quite seem to believe that the results reported in this paper is significant enough. One can refer to the following paper for a few relatively new semi-supervised learning approaches:

https://arxiv.org/pdf/1804.09170.pdf

Table 4 is more baffling and less convincing. Because deep networks are volatile, it is hard to show this kind of result and hope people will be convinced. I would rather the author has trained both approaches to completion and then compare the end result. Also we still need comparisons with state-of-the-art semi-supervised learning approaches.

---

### Official Review · AnonReviewer3 · 2018-11-05
**Intresting work, but the effectiveness of the proposed method is not validated**

**Rating:** 4
**Confidence:** 3

**Review:**

The idea of calculating a score to indicate the usefulness of a sample for training deep networks by analyzing the neural activations in semi-supervised learning is interesting.

However, the effectiveness of the proposed method is not validated. In the cifar-10 semi-supervised image classification experiment, other semi-supervised learning methods are not compared. In my experiments, simply applying the trained model in the labeled data to obtain pseudo labels on the unlabeled data can obtain significant improvements.

Theoretically, the proposed scoring method uses the pre-trained model to obtain correct activations that has two problems: (1) The evolving power that may be produced by the unlabeled data is constrained. (2) If there are a few numbers of labeled examples, it is very hard to learn a network; thus, the correct activation is not reliable.

---

### Meta-Review · Area_Chair1 · 2018-12-13
**Clear ratings of reject from reviewers**

**Confidence:** 4
**Recommendation:** Reject

**Metareview:**

Reviewers are in full agreement for rejection.